


# Dust emission in farmland caused by aerodynamic entrainment and surface renewal

Hongchao Dun[1], Ning Huang[1]

[1]Key Laboratory of Mechanics on Disaster and Environment in Western China, Lanzhou University, Lanzhou, 730000,
China

*Correspondence to*: Ning Huang (huangn@lzu.edu.cn)

**Abstract.** Mineral dust emission is an important physical process related to gas-particle interaction and soil moisture temporal change, which is crucial on global circulation patterns and biogeochemical cycles. The existing dust models are usually semi-empirical functions and far from reliable prediction of dust emission rate ignoring the different phases in dust emission. Here, a dust emission model combined with aerodynamic entrainment and surface renewal mechanisms is established to simulate the physical phased dust emission in farmland. We use a soil moisture transport module to simulate the temporal soil surface moisture, and a simple and feasible scheme to calculate the amount of free grains exposed on soil surface. The model reproduces three typical phases of dust emission: aerodynamic entrainment, dry soil saltation and surface renewal, in which soil moisture is the dominating limiting factor and the dust emission rate remains low. Results show that our model is an effective method to predict the dust emission rate.

## 1 Introduction

Mineral dust emission caused by wind erosion is a primary component of the global dust cycle in our Earth system and a major factor of desertification (Joussaume, 1990), due to the loss of nutrient rich fine particles, coarsening of topsoil, decreasing of soil fertility and declining of land productivity (Mahowald, 2011; Huang et al., 2012). Although saltation bombardment and aggregates disintegration have been proposed as important mechanisms of dust emission (Marticorena and Bergametti, 1995; Herrmann and Parteli, 2007; Kok and Renno, 2008; Shao, 2004, 2008; Carneiro et al., 2013; Pähtz et al., 2013; Újvári et al., 2016), they are still not fully understood and existing dust models are far from reliable prediction of dust emission rate (Webb and Strong, 2011; Evan et al., 2014; Dupont et al., 2015). Recently, it has been found that the contribution of direct aerodynamic dust entrainment is substantial in nature (Macpherson et al., 2008; Shao, 2008; Sow et al., 2009; Klose et al., 2014), which leads Zhang et al. (2016) to studies on different phases in the dust emission. Their results indicated that, in the initial phase of dust emission from a natural soil surface, aerodynamic entrainment should be the dominant mechanism and dust might be supplied by free grains exposed on soil surface. As the free grains were eroded, aerodynamic entrainment was less efficient and soil particle saltation became the main pathway to maintain dust emission (Fig. 1a).





When dust emission proceeds to the phase mainly driven by saltation, soil particles are gradually denuded away and the surface height is reduced, which exposes the underlying soil particles with higher internal moisture content to the surface. This increased soil moisture is a critical to inhibit the saltation by changing initial soil movement due to the wind (Chen et al, 1996) (Fig. 1b). The new wetter layer will be removed once dry enough by wind and this phenomenon will repeat itself, i.e.,

a surface renewal process due to soil moisture change takes place. Such phenomenon is common in natural, but no attempts have been made to model this process in physical sense coupling evaporation and surface renewal (Cornelis and Gabriels, 2010). Generally, in dust emission models, the soil moisture in whole topmost layer from regional or global land surface is considered to be constant during a dust erosion event, which leads to an underestimation of simulated dust emission (Bergametti et al., 2016; Xin and Sokolik, 2015), because soil surface is dried rapidly and a moisture gradient is formed in

the topmost layer caused by intense evaporation in arid and semiarid region (Bisal and Hsieh, 1996; Webb and Strong, 2011) (Fig. 1d).

In this paper, three phases of dust emission in wind erosion events will be included as (i) aerodynamic entrainment, (ii) saltation transport and (iii) surface renewal caused by soil moisture. Our model contains the amount of free grains exposed on soil surface and the temporal soil surface moisture (Fig. 1c). According to a saltation flux model of drifting soil, an

efficient method will be established to predict natural dust emission with significant temporal heterogeneity under nature conditions considering the aerodynamic entrainment and surface renewal mechanisms (AESR).

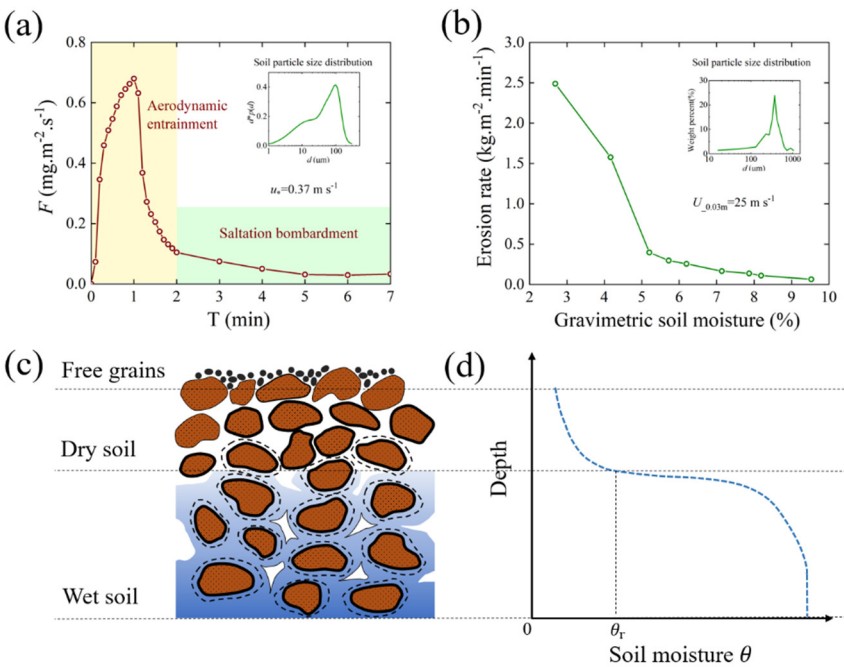

Figure 1. (a) The initial phase of dust emission due to aerodynamic entrainment and the second phase due to soil saltation (Zhang

et al., 2016). (b) Soil moisture initials the erosion and dust emission rate in the third phase (Chen et al, 1996). (c) Illustration of the



soil structure: (i) free grains for aerodynamic entrainment, (ii) dry soil layer for soil saltation, and (iii) wet soil that inhibits the saltation and dust emission. (d) Soil moisture distribution along the depth and the residual soil moisture content ($\theta_r$) is the threshold between dry and wet soil.

## 2 Dust emission model

### 2.1 Aerodynamic entrainment

Opposed to the assumption that soil grain size distribution is invariant in vertical direction of soil, many study results indicate that there is a thin layer composed of free fine dust grains on soil surface (Shao, 2008; Zhang et al., 2016). Under the action of strong solar radiation, soil aggregates in the thin layer of soil surface absorb lots of energy, the inter-grain water is completely drained and there are no enough cohesive forces to maintain the aggregate morphology, which result in the soil aggregates disintegrate to free fine dust grains. Due to the shielding effect, solar radiation cannot penetrate the topsoil, thus the thickness of the free dust layer can be considered as the average soil grain radius. The internal structure of free dust layer can be simplified as the schema shown in Fig. 1c.

Thus, the ratio of the area covered by free dust to the total area in vertical direction can be calculated as:

$$f_{dust} = 1 - \frac{\sqrt{R^2-(R-z)^2}}{R} \quad 0 < z < R, \tag{1}$$

where $R$ is average radius for soil aggregates be exclusive of free fine dust grains on soil surface, z is vertical depth away from soil surface.

$$F_{dust} = f_{dust} \cdot D_* \cdot u_*^n \left(1 - \frac{u_{*_t}}{u_*}\right), \tag{2}$$

where $u_*$ is the friction velocity of wind, $u_{*_t}$ is the threshold friction velocity of wind in the initial soil movement, and $D_*$ is the coefficient obtained from experiments.

### 2.2 Surface renewal process

Since saltation transport is the main mechanism in dry and wet soil layers, a widely used formula for horizontal saltation flux is given as Equation (3) according to Owen (1964),

$$Q(d_s) = c_0 \frac{\rho_a}{g} u_*^3 \left(1 - \frac{u_{*_{wt}}^2(d_s)}{u_*^2}\right), \tag{3}$$

where $d_s$ is the grain size of saltate soil, $c_0 = 0.25 + v_t / (3u_*)$ and $v_t = 1.66(\sigma_\varphi gd)^{1/2}$ as the grain terminal velocity, $\sigma_\varphi$ the density ratio of grain to air, $u_{*_t} = \sqrt{A_N\left(\left(\rho_p - \rho\right)gd / \rho + \gamma / \left(\rho d\right)\right)}$ is the threshold friction velocity (Shao and Lu, 2000),


$u_{*wt} = u_{*t} + 7.5\theta \rho_w / \rho_s$ is the threshold friction velocity considering soil moisture (Horikawa et al., 1983), and $\theta$ is the volume moisture content (%).

We assume that soil grain size distribution is invariant in soil under the free dust layer, and the grain size distribution of natural soil can be fitted with an overlay of multiple lognormal formula (Zhang et al., 2016):

$$p(d_s) = \frac{1}{d} \sum_{j=1}^{N} \frac{w_j}{\sqrt{2\pi}\sigma_j} exp\left(-\frac{(lnd-lnD_j)^2}{2\sigma_j^2}\right), \tag{4}$$

where $N$ is modes number of the superimposed lognormal distribution, the maximum is 4. $D_j$ and $\sigma_j$ are median mass grain size and geometric standard deviation of the $j$th grain size distribution mode in lognormal distribution. $w_j$ is weight ratio of $j$th grain size distribution mode.

Based on the horizontal saltation flux $Q(d_s)$, the dust emission rate by saltation can be obtained as Zhang et al. (2016),

$$F_{sal} = \eta \cdot \exp(\varphi \cdot u_*) \cdot Q(d_s), \tag{5}$$

where $\eta$ and $\varphi$ are coefficients.

In the surface renewal process, free grains or soil particles are gradually denuded away by wind, and the surface height decrease can be expressed as,

$$\Delta z = \Delta Q / \rho, \tag{6}$$

where $\Delta Q$ is the mass loss and $\rho$ is the density of free grain or soil particle layers.

Although the water content in dry soil is too low to satisfy the requirement of continuous medium, it can still influence the initial movement of soil particles (Ravi et al., 2006). Then, the soil moisture distribution driven by water evaporation from wet soil layer through the dry soil layer to atmosphere can be calculated as (Cass et al., 1984),

$$\frac{\partial C}{\partial t} = D_v \frac{\partial^2 C}{\partial z^2}, \tag{7}$$

where $C$ is the vapor concentration, $D_v$ is the diffusion coefficient, and the relationship of vapor concentration and soil moisture on soil particles surface can be expressed as (Ajaev, 2012),

$$C = e^{-\lambda/\theta^3}, \tag{8}$$

### 2.3 Evaporation process

The unsaturated soil hydrodynamics formula for water movement in wet soil is (Richards, 1931),

$$\frac{\partial \theta}{\partial t} = \frac{\partial}{\partial z}\left(D(\theta)\frac{\partial \theta}{\partial z}\right) + \frac{\partial K(\theta)}{\partial t}, \tag{9}$$

where $K(\theta)$ is the soil hydraulic conductivity and $D(\theta)$ is the hydraulic diffusivity. According to Van Genuchten (1980), empirical formulas of $K(\theta)$ and $D(\theta)$ are,



$$K(\theta) = K_s \Theta^{0.5} \left[ 1 - \left( 1 - \Theta^{1/m} \right)^m \right]^2, \tag{10}$$

$$D(\theta) = \frac{(1-m)K_s}{am(\theta_s - \theta_r)} \Theta^{0.5 - 1/m} \left[ \left( 1 - \Theta^{1/m} \right)^{-m} + \left( 1 - \Theta^{1/m} \right)^m - 2 \right], \tag{11}$$

where $K_s$ is the saturated hydraulic conductivity of soil, $\Theta = (\theta - \theta_r)/(\theta_s - \theta_r)$ is the relative saturation, $m$ is the soil

property parameter, $\theta_r$ is the residual soil moisture content and the threshold between dry and wet soil, and $\theta_s$ is the

saturated soil moisture content. Since the wind velocity $u$ is the principal factor, the evaporation rate $E$ can be expressed as

(Schmutz and Namikas, 2018),

$$\frac{E}{E_0} = \begin{cases} a\theta + b, & \theta > \theta_r \\ 1/(c + d\delta), & \theta \leq \theta_r \end{cases} \tag{12}$$

where $E_0 = (e_0 - e_z) \cdot (\alpha + \beta u)$ is the evaporation rate on water surface (Ta et al., 2009), $e_0$ is the saturated vapor pressure in

a thin layer above the pure water surface, $e_z$ is the vapor pressure at height z above the water surface, and $\delta$ is the thickness

of dry soil and determined by $\theta_r$.

## 2.4 Calculation procedures

The simulations for dynamic dust emission processes are carried out according to the following procedures:

1.    establish the governing equations of dynamic dust emission model;
2.    set the initial boundary conditions according to the simulation examples;
3.    mesh the grids for computational domain;
4.    calculate the evaporation rate $E$ with Eq. (12), then get the soil moisture distribution at this time step end with Eq. (7),

     (8), (9), (10), (11); calculate the average of soil surface moisture at the begin and end of this time step;
5.    according the Eq. (3), (4) to simulate the horizontal saltation flux $Q(d_s)$, and get the soil erosion depth and the new soil

     surface position at the end of this time step with Eq. (6);
6.    calculate the dust emission rate $F$ with Eq. (1), (2), (5);
7.    get the information of soil surface position and moisture content at this time step end from the results of procedures 4-6;
8.    repeat procedures 4-7 until satisfied the simulation time demand.

## 3 Results and analysis

Typically, before a dust emission or wind erosion event, a continuous soil drying process usually already exits to increase its

erodibility (Webb and Strong, 2011). Therefore, we established a dust emission model including aerodynamic entrainment

and surface renewal mechanisms. We calculated a 10-day evaporation process without wind from a soil with a moisture





content of 0.025, and rebuilt the erodible soil structure containing dry layer and wet layer in nature. All the dust emission and wind erosion simulation results in this study were based on the soil initial conditions, and the soil moisture distribution is shown in Fig. 2.

## 3.1 Temporal changes for soil moisture distribution and surface position

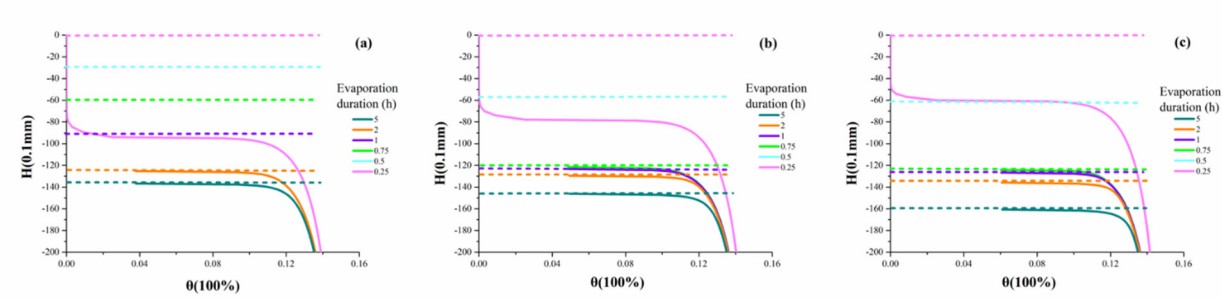

**Figure 2: Temporal changes for soil moisture distribution and surface position with different friction velocity $u_*$: (a) $u_* = 0.4 m/s$; (b) $u_* = 0.45 m/s$; (c) $u_* = 0.5 m/s$. Solid lines are soil moisture distributions; dotted lines are new soil surface positions and the soil above these lines is denuded away.**

As shown in Fig. 2a, the distribution of moisture had little change in the first hour of wind erosion, but the surface height kept decreasing. At this phase, the dry soil layer was mainly denuded by wind. Because the soil moisture content inside was low and the resistance to wind erosion was weak, the rate of surface decline is relatively high. In the second hour, the dry soil layer had disappeared, and the wet soil layer was exposed and denuded by wind as the surface layer with a moisture content of about 0.04. In this process, air flow accelerates the evaporation process, and the surface moisture content was gradually decreased, thus soil grains could be driven by wind and form a new soil denudation process. Then, the system achieved a new dynamic balance.

In the cases with high wind velocity as shown in Figs. 2b and 2c, the increased wind velocity significantly enhanced the soil denudation process, and the period of dry soil erosion was significantly shortened. The surface moisture content remained stable and was slightly lower than 0.05. Along the wind erosion, wet soil layer was gradually denuded away and the surface height was decreased. With a high wind velocity, the erosion effect on dry soil layer could hardly be improved, while the ability of erosion on wet soil layer still had great potential.





## 3.2 Temporal changes for evaporation and soil structure

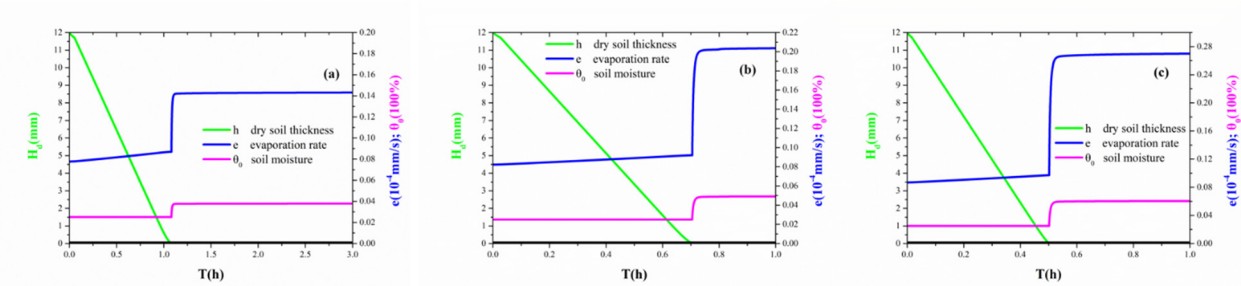

**Figure 3: Temporal changes for evaporation and soil structure with different friction velocity $u_*$: (a) $u_* = 0.4 m/s$; (b) $u_* = 0.45 m/s$; (c) $u_* = 0.5 m/s$. Green lines are dry soil layer thicknesses; black lines are the decrease velocity of dry soil layer thicknesses; blue lines are the evaporation rates; pink lines are the soil moisture on wet layer surface, which determine the evaporation rates.**

Fig. 3 shows the temporal changes in evaporation and soil structure with friction velocity $u_*$. Results showed that the processes of soil wind erosion and evaporation proceed simultaneously. During the wind erosion and dust emission of soil, the erosion process of free dust occurred first. Due to the small size of free dust grains and their stronger inter-grain cohesion than lager soil grains, the surface denudation rate was slightly lower. With the depletion of free dust, supply limits begin to form, and the object driven by wind changed from free dust to large soil grains, indicating the erosion process of dry soil layer began. In this process, due to the increase of grain size and the partial cohesion provided by inter-grain water, the wind erosion process was partially inhibited, and the surface decline rate remained stable. After the dry soil layer was consumed, it immediately turned to the erosion phase of the wet soil layer. Because the transition between the two phases was very rapid, the evaporation and erosion processes of soil quickly reached to new dynamic balances, in which the moisture content on soil surface became the main limiting factor for soil wind erosion.

The increase of wind velocity enhanced the erosion rate of dry and wet soil at the same time. On the other hand, it improved the surface moisture content in the new dynamic equilibrium. When the friction wind velocity $u_*$ was 0.40m/s , 0.45m/s and 0.50m/s , the corresponding surface moisture content was about 0.04, 0.05 and 0.06 with a linear ship change.





## 3.3 Three main phases in dust emission process

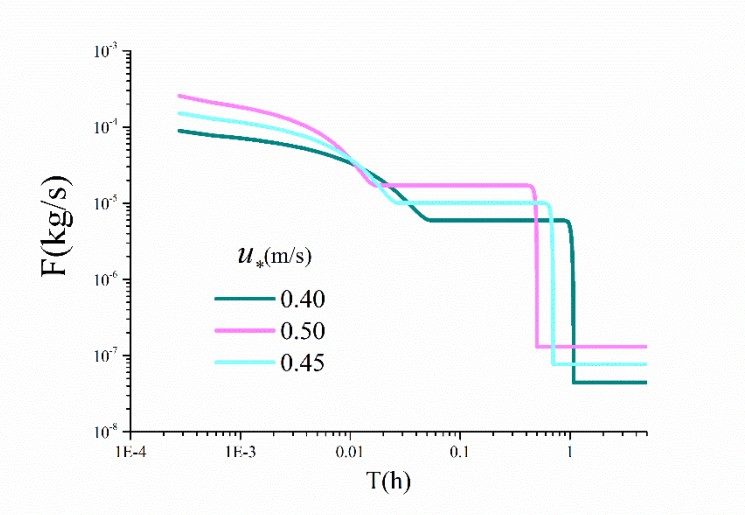

**Figure 4: Three main phases in dust emission process with different friction velocity $u_*$: (i) aerodynamic entrainment is the primary mechanism in first phase, and the dust emission rate decreases rapidly in a few minutes, (ii) saltation transport is the main mechanism in the second phase, and the dust emission rate maintains at a relatively high level, (iii) soil moisture becomes the dominating limit factor in the third phase, and forms little dust emission.**

Fig. 4 shows different phases in the dynamic dust emission process. During the dust dynamic emission, the dust emission rate curve under different wind velocities showed a similar change trend, which could be divided into three main emission phases. The first phase was supplied by free fine dust mainly and aerodynamic entrainment emission was the primary mechanism. Due to the smaller grain size of free dust and the lower cohesive forces reduced by soil aggregates, the dust emission rate was very high in this phase. However, because the uneven distribution of free dust content in the vertical direction, the dust emission rate in this phase was decreased rapidly with time, reflecting the supply limitation of free dust. While the free dust layer was consumed by wind erosion, saltation transport became the main mechanism in this phase. Because dust emission from big grains was relatively high and erosion processes were restrained accordingly in this phase, the dust emission rates were decreased significantly compared with that in the first phase. Therefore, the thickness of dry soil layer was main limiting factor of the dust emission in this phase. After the dry soil layer disappeared, the dust emission turned into the third phase, in which wet soil was the limit factor and saltation transport was the main mechanism. The existence of water between the soil grains hindered the releasing process of wind erosion and further reduced the dust emission rate. In this phase, soil moisture content became the main limiting factor of dust emission rate.

### 3.4 Simulation for a field dust emission event

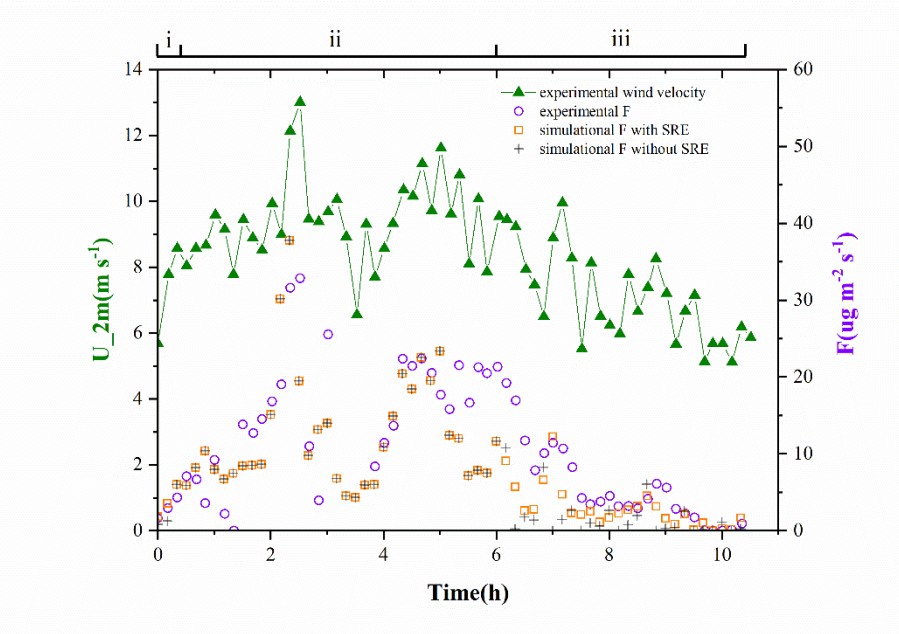

**Figure 5: Simulation results of a field dust emission event. Green triangles are wind velocity data measured at the height of 2 m; purple hollow rings are the measured air dust emission rate. Orange hollow squares are the simulated dust emission rate considering AESR, it is consistent with the experimental results in all three phases. Gray hollow crosses are the simulated dust counts without AESR, the effects of aerodynamic entrainment (phase i) and soil moisture (phase iii) cannot be embodied.**

In this study, a simulation on a field dust emission event had been done and the results were showed in Fig. 5. The results were consistent with that based on a dust emission event lasting for 11 hours in a field with natural wind from Kjelgaard et al. (2010), which verified the accuracy of the model in this study. Due to the existence of free dust on the soil surface, the dust emission rate in phase i was relatively high, and the dust concentration in the air was increased rapidly. This model simulated the dust emission process caused by aerodynamic entrainment in nature for the first time considering AESR. Limited by the supply of free dust in phase ii, saltation transport became the main mechanism and the dust emission rate started to decrease rapidly after, marking the end of the aerodynamic entrainment emission phase caused by wind. Through the simulations reflected that the dust emission rates, whether considering AESR or not, had slight differences. In phase iii, surface renewal caused by soil moisture became the main mechanism, the saltation transport and dust emission efficiency were greatly reduced. Even the wind velocity was very low, the airborne dust concentration could still be maintained at a low level due to AESR which dried the surface wet soil and supplied the soil erosion and dust emission.



## 4 Conclusions

In this paper, we analyzed the mechanism of dust emission process and described a new model to simulate the dust emission process. This model included temporal changes of soil moisture and surface renewal processes, which were principal influence factors of soil erosion. During a dust emission and wind erosion event, the soil was gradually denuded away and
the surface height reduced, which exposed the underlying soil particles with higher internal moisture content to the surface. The surface renewal increased the soil surface moisture. On the other hand, the evaporation process eliminated the water in soil and decreased soil surface moisture. Therefore, evaporation and surface renewal processes made up a feedback system, which could reach dynamic balance conditions in different phases.

Furthermore, the dust emission process could be divided into three main emission phases. The first phase was that particle
emission was supplied by free fine dust mainly with aerodynamic entrainment as the primary mechanism. And the dust emission rate was very high in this phase. The dust emission rate was then decreased rapidly along the time due to the supply limitation of free dust. While the free dust layer had been consumed, saltation transport becomes to main mechanism in the second phase. At this stage, dust emission rate was decreased significantly, and dry soil layer thickness was the main limiting factor. When the dry soil layer disappeared, dust emission turned into the third phase, in which wet soil was the main supply
and saltation transport was the main mechanism. The existence of water between the soil grains hindered the releasing process of wind erosion and further reduced the dust emission rate, and soil moisture content became the main limiting factor of dust emission rate.

To our best knowledge, this study could provide a comprehensive model for temporal dust emission process with more physical mechanisms to describe the actual emission event in nature.

**Acknowledgements and data availability**

Thanks to the support of National Natural Science Foundation of China (41931179, 11772143). The underlying data can be found online at: https://doi.org/10.6084/m9.figshare.8246345.v1.

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
