# Peer review of "Dust emission in farmland caused by aerodynamic entrainment and surface renewal"

_Atmospheric Chemistry and Physics, 2020_

## Referee Comment (RC1) · Anonymous Referee #1 · 15 Jan 2021

Zhang et al., (2016) had put forth the renewal mechanism of fine particles in a soil's top layer, which they think is critical to sustaining dust emission. The work Dun and Huang presented here clearly attempts to build on that study simulating the dust emission process in farmland using a dust emission model with combined aerodynamic entrainment and surface renewal mechanisms previously proposed. They are trying to show that their model is effective to predict dust emission in farmland. In general, however, I think the performed approach and methodology are subject to major deficiencies, and the results are questionable. In many places, the statements drawn by the authors lack sufficient evidence: the readers would appreciate it if the authors could explain some crucial aspects in detail. Some sections also needed to be restructured. So, I regret

that I am unable to recommend publication of this manuscript in its present form in ACP.

P – Page; L – Line (please use continuous line numbering instead of restarting numbering on every page. The current line numbering makes the review process painful!)

Major comments:

As the authors themselves pointed out, the simulated dust emission rate only slightly differs between including and excluding the aerodynamic entrainment and surface renewal mechanisms. I cannot find clear evidence supporting the main conclusion of this study. I have no idea based on what the authors concluded that "the model is an effective method to predict the dust emission rate". I encourage the authors to try to improve the model results, or, if that proves impossible, then learn why the model is not working and write a thoughtful and candid report characterizing the issues and the lessons learned by the attempt. But currently, I am really struggling to find out the scientific merit of this work.

Also, the evaluation of model performance (Section 3.4) relies on only one dust event, and there is no detailed quantitative analysis of the modeled and experimental data. Strictly speaking, a more intensive evaluation is required to put the conclusion on a more solid statistical basis. Event for comparison between the simulated and measured dust flux at the current level, the discussion seems somewhat subjective. The authors divided the field dust event process into three main phases without any justification. What kind of data or sensitivity study is there showing that this kind of phase division is reasonable? How did the authors distinguish the contribution of the dust emission from different mechanisms in each phase? How did they attribute the primary emission mechanism in the first phase to aerodynamic entrainment? Besides, the statement in the model evaluation section is a little bit not more candid. I would not say that the dust emission rate in the first phase, according to the authors' division, is high (the "relatively high" is vague): it is much lower than peak values registered in the second phase. In that short section, the authors mentioned the dust concentration twice. But

I did not see any dust concentration data presented in the manuscript for this event to support those statements.

Also related to the field dust emission event. The authors should have to state where they get the data and how the experiment was set up to obtain the wind velocity and dust emission flux.

The "farmland" only appears in the title and the abstract but is not mentioned anywhere else, which looks weird. The authors need to introduce somewhere in the manuscript the unique property of farmland surface from the dust emission perspective and make it clear why the model presented here is suitable for use to model the dust emission in farmland. It would be more interesting if the authors could quantitatively quantify how big the difference would be on the dust emission rate with and without the surface renew by soil moisture.

The calculation procedure is not very clear to me. I think there would be an update of theta after obtaining the evaporate rate. So, the theta in Eq. 12 is actually at a time step right before the current one. The authors may want to clarify the different time steps the theta is at in the equations to avoid any possible confusion. A flow chart would be helpful too.

I would encourage the authors to construct more sensitivity tests on some key parameters that control the soil moisture prediction to see quantitively how they affect the dust emission rate in farmland.

Many variables are using in the equations without any definition. Values for constant parameters used in the model are also missing (please see detailed comments below). I would encourage the authors to specify those constants such that readers can tell if they are within the reasonable range and reproduce the results.

Minor comments:

P1; L18: please give the size range for "fine particle".

P1; L25-27: "Their results indicated that, in the initial phase of dust emission from a natural soil surface, aerodynamic entrainment should be the dominant mechanism and dust might be supplied by free grains exposed on soil surface." This statement seems not really correct. The aerodynamic entrainment could be crucial for dust emission only under certain circumstances. Here, I think that the authors exaggerated the importance of aerodynamic entrainment to dust emission.

P2; L7: how thick the topmost layer is defined?

P3; Section 2: the readers would appreciate a few sentences right after the section heading to explain how Section 2 is organized before diving into the subsections.

P3; Eq. 1: consider adding a plot to Figure 1d to illustrate the vertical profile of the free fine dust coverage. Also, labeling the thickness of the free dust layer in Figure 1c would help readers get the point readily.

P3; Eq. 2: please introduce Fdust in the main text before showing this equation. Also, what is the value of "n" used in this study?

P3: does d in Eq. 3 refer to diameter? What is the size limit in the dust model?

P3; L25: please introduce U*wt first and then U*t. Also, please define AN and gamma, and specify the constants used in the dust model.

P4; L1: please define rou w and rou s.

P4; Eq. 4: what' the difference between ds and d? ds is the soil grain size, then what does d stand for? It seems to be a typo, as it does not make sense to have d on the right-hand side but ds on the other, and d only comes into play in this lognormal formula. Also, should specify how many modes ("N") and the lognormal distribution parameters used for calculation.

P4; L7: considering deleting "in lognormal distribution". This term seems redundant, considering it had been mentioned in the sentence right before. No need to repeat the

information.

P4; L16: Text starting from this line within this subsection is talking about soil moisture distribution, separated from the content of this section. Since the soil moisture distribution is strongly affected by the evaporation rate (in Section 2.4, the authors also cite Eqs. 7 and 8 together with Eq. 9 and so on), it would be better to put it in Section 2.3, where the authors detailed how to predict the soil moisture content. With this adjustment, the authors may want to change the subtitle of 2.3 to "Soil moisture distribution" or any other similar.

P4; L20: specify the value for Dv.

P4; Eq. 8: define lambda and specify the value used.

P5; Eq. 10, 11, and 12: specify Ks, m, a, b, c, and d etc.

P5; Section 2.4: please provide more details about each step: how the initial boundary conditions were set; what's the grid resolution; what's the time step etc. Please specify. Also, see major comments on step 4.

P5; L24-26: is there any cause-and-effect relationship between the two sentences? Please explain.

P5; L26 to P6; L2: sentence difficult to follow. Why a 10-day evaporation process? Is the "soil initial condition" referring to the one right after the 10-day evaporation finished? But under which friction velocity was used as you have three in Fig. 2abc?

P6; L20-21: unclear sentence. What does it mean when saying that the "erosion effect on dry soil layer could hardly be improved"?

P6; Fig. 2: why in the first 0.25 hr, the surface position with $U^*$=0.5 m/s is higher than with $U^*$=0.45 m/s? Why in the first hour the soil moisture content at the newly exposed surface with $U^*$=0.5 m/s is higher than with $U^*$=0.45 m/s, even though the surface position is comparable between the two cases? Another interesting but missing point

is that increasing U* from 0.45 to 0.5 m/s did not lower the surface position in the first 0.5 and 0.75 hrs as much as increasing U* from 0.40 to 0.45 m/s.

P7; Fig. 3: I think it could be interesting to also show the total soil thickness. Fig. 3s: I did not see black lines.

---

## Referee Comment (RC2) · Anonymous Referee #2 · 17 Jan 2021

This paper presents a theoretical modeling study of dust emission from aerodynamic entrainment and saltation including an implementation of the surface renewal mechanism. Specifically, a parameterization of the free dust layer and a soil moisture transport module are developed and incorporated. The model simulated dust emission rates are compared with the observations from a field study.

It is an interesting modeling study, as it illustrates the time evolution of dust emission rates on the process level, governed by the ambient conditions such as surface wind speeds and soil moisture. The effects of wind erosion and soil moisture changes due to evaporation are modeled in both aerodynamic entrainment and saltation processes.

[Figure]

While the quantitative results may depend on the model specifications, it characterizes the relative importance and temporal dependence of the surface wind and soil properties in dust emission processes.

However, the manuscript needs major revisions in model description and evaluation before it could be considered for publication. There are two major concerns. First, a main contribution of this work is the development of this process model for dust emission. But the discussions about the model formulation and uncertainties in parameters are insufficient (detailed below in specific comments), making it difficult to determine if the results/conclusions are reasonable and where the model is applicable (or not). Further, the model evaluation includes one case study only comparing the simulated dust emission fluxes with a dust experiment. And the analysis of the model-data comparison is ad hoc and insubstantial.

Specific comments are given below: (1) The parameterization of the free dust area in Equation (1) is introduced the first time by this study. It is not justified how it is formulated: is it physically based or empirically fitting based on the experimental data? The equation implies a sharp decrease in available free dust fraction close to the surface. Since the predicted changes of dust emissions due to the aerodynamic entrainment is sensitive to the function, verification of the predicted free dust area with the experimental data or theoretical justification is necessary. (2) Also, in Equation (1), it is unclear what the R value is used for the radius of free dust grains and how it is determined; and is this parameter variable, depending on the surface type? How does this equation relate to the results in Section 3.1 and Section 3.2, Figures 2 and 3, i.e., is the dry soil thickness (H_d) sensitive to R in Equation (1)? (3) Equation (10) and (11): what is the definition of m and what is its typical value? (4) Equation (12): is the calculation of theta and evaporation rate applicable only over the wet soil? If the fraction of dry soil is > 0, i.e.,f_dust in Equation (1), will the theta and evaporation rate be calculated for that layer and how? (5) Section 2.4: a flow diagram would help illustrate the procedure. Lots of the detail about the model experiment are omitted. As mentioned in

the main comment above, without those detail it is difficult to determine whether the results are reasonable. For instance, what is the initial soil moisture profile used? Is it representative for farmland, which seems to be the land surface type of interest? The model domain is unclear: is it a 1-D or 3-D model? What is the model horizonal and vertical resolution? Are there any horizonal variability in the initial conditions of soil moisture content and surface winds? (6) Figure 3: there is no black lines plotted in any of the panels (a)-(c). During the first hour when H_d>0, why the soil moisture remains constant but there is a slow increase in evaporation rate? is the stepwise increase in evaporation rate and soil moisture related to the initial soil moisture profile assumed? (7) Figure 4: in order to attribute the dust emission flux to a primary mechanism, it would make sense to plot the contribution of aerodynamic entrainment separately from that due to saltation transport. Sensitivity studies of other important parameters in the model such as soil moisture profile and surface air temperature/humidity would help in strengthening the findings from the model simulations. (8) Section 3.4: this model evaluation section needs to be expanded. As mentioned in the main comment, it is unclear if the model configuration is comparable to the experimental conditions such as soil type, moisture content and profile. More quantitative analysis of the model-data differences is needed, for instance, in terms of RMSE, correlation, or other statistical measures. The impact due to Surface Renewal and Evaporation (SRE) is visible only after 6 hours; however, this seems to be inconsistent with the model simulations in Section 3.1-3.3, which show that the SRE effects occur within $\sim$ 1hr. Why is that? Even after including the SRE effect, the differences between the model and observations are still quite large. It is not very convincing that the developed model captures the time evolution of the dust emission fluxes effectively in this case study. Furthermore, to attribute the changes of dust emission to a specific process: aerodynamic entrainment of free dust or saltation transport due to wind erosion, it is necessary to decompose the predicted dust emission fluxes by process. (9) Page 9, lines 14-15: dust emission caused by aerodynamic entrainment has been demonstrated in a number of previous studies such as Klose and Shao (2012) and Zhang et al. (2016). For the statement

"this model simulated the dust emission process caused by aerodynamic entrainment in nature for the first time", clarification about how this study is different from previous studies on this process is needed.

Overall, the manuscript is an interesting modeling study of dust emission processes based on the theoretical understanding. However, it requires significant improvement and justification in model description and evaluation, in order to support the findings of their model simulations.

Please also note the supplement to this comment:
https://acp.copernicus.org/preprints/acp-2020-1021/acp-2020-1021-RC2-supplement.pdf

---

## Author Comment (AC1) · 16 Mar 2021

This paper presents a theoretical modeling study of dust emission from aerodynamic entrainment and saltation including an implementation of the surface renewal mechanism. Specifically, a parameterization of the free dust layer and a soil moisture transport module are developed and incorporated. The model simulated dust emission rates are compared with the observations from a field study. It is an interesting modeling study, as it illustrates the time evolution of dust emission rates on the process level, governed by the ambient conditions such as surface wind speeds and soil moisture. The effects of wind erosion and soil moisture changes due to evaporation are

modeled in both aerodynamic entrainment and saltation processes. While the quantitative results may depend on the model specifications, it characterizes the relative importance and temporal dependence of the surface wind and soil properties in dust emission processes. However, the manuscript needs major revisions in model description and evaluation before it could be considered for publication. There are two major concerns. First, a main contribution of this work is the development of this process model for dust emission. But the discussions about the model formulation and uncertainties in parameters are insufficient (detailed below in specific comments), making it difficult to determine if the results/conclusions are reasonable and where the model is applicable (or not). Further, the model evaluation includes one case study only comparing the simulated dust emission fluxes with a dust experiment. And the analysis of the model-data comparison is ad hoc and insubstantial.

Response: Thanks for the positive comments and useful suggestions. According to the reviewer's suggestions, we had discussed more about the model formulation and uncertainties in parameters, provided more data and explanations, and improved the quality of whole manuscript. Please see the responses to the following questions.

Specific comments are given below: 1. The parameterization of the free dust area in Equation (1) is introduced the first time by this study. It is not justified how it is formulated: is it physically based or empirically fitting based on the experimental data? The equation implies a sharp decrease in available free dust fraction close to the surface. Since the predicted changes of dust emissions due to the aerodynamic entrainment is sensitive to the function, verification of the predicted free dust area with the experimental data or theoretical justification is necessary.

Response: Thanks for the comment and suggestion. The parameterization of the free dust area in Equation (1) is physically based. According to the reviewer's suggestions, we added the sentence "we simplify the soil aggregate particles to spheres, and the free fine dust grains are filled in the particle gaps" in lines 69 of the revised manuscript. We added the verification of the predicted free dust area with the experimental data

in lines 245-271, in which the aerodynamic entrainment caused by fine dust is accord with the wind tunnel experiments by Zhang et al. (2016):

Figure 5: Sensitivity of dust emission flux F to friction velocity, specific humidity and initial soil moisture content. Three main phases in dust emission process: (i) aerodynamic entrainment is the primary mechanism in first phase, and the dust emission rate decreases rapidly in a few minutes, (ii) saltation transport is the main mechanism in the second phase, and the dust emission rate maintains at a relatively high level, (iii) soil moisture becomes the dominating limit factor in the third phase, and forms little dust emission.

Fig. 5 shows different phases in the dynamic dust emission process. During the dust dynamic emission, the dust emission rate curve under different wind velocities showed a similar change trend, which could be divided into three main emission phases. The first phase was supplied by free fine dust mainly and aerodynamic entrainment emission was the primary mechanism. Due to the smaller grain size of free dust and the lower cohesive forces reduced by soil aggregates, the dust emission rate was very high in this phase. However, because the uneven distribution of free dust content in the vertical direction, the dust emission rate in this phase was decreased rapidly with time, reflecting the supply limitation of free dust. Klose and Shao (2012) study the aerodynamic entrainment in the absence of saltation as large eddies intermittently produce strong shear stresses on the surface and entrain dust particles into the air, in which convective atmospheric condition is major influence factor rather than soil property. While the free dust layer was consumed by wind erosion, saltation transport became the main mechanism in this phase. Because dust emission from big grains was relatively high and erosion processes were restrained accordingly in this phase, the dust emission rates were decreased significantly compared with that in the first phase. Therefore, the thickness of dry soil layer was main limiting factor of the dust emission in this phase. After the dry soil layer disappeared, the dust emission turned into the third phase, in which wet soil was the limit factor and saltation transport was the main mechanism.

The existence of water between the soil grains hindered the releasing process of wind erosion and further reduced the dust emission rate. In this phase, soil moisture content became the main limiting factor of dust emission rate. Zhang et al. (2016) study the first two phases in wind tunnel experiments, but the third phase is difficult be rebuilt due to the time limit for wind tunnel operation. Tests are performed to investigate the dependency of dust emission F on friction velocity, specific humidity and initial soil moisture content. For constant friction velocity and initial soil moisture content, F has a small difference with large specific humidity, and clearly increases for in small specific humidity. Figure 5c also shows that final dust emission rate F is insensitivity with initial soil moisture content.

2. Also, in Equation (1), it is unclear what the R value is used for the radius of free dust grains and how it is determined; and is this parameter variable, depending on the surface type? How does this equation relate to the results in Section 3.1 and Section 3.2, Figures 2 and 3, i.e., is the dry soil thickness ($H\_d$) sensitive to R in Equation (1)?

Response: Thanks for the comment. R is average radius for soil aggregates and determined by the soil grain size distribution. We modified this part in lines 71: "where R is the highest proportion radius of soil particle size distribution". R mainly affects the dust emission rate in aerodynamic entrainment process, and Figure 4 is a good example. Soil moisture distribution and soil structure in Figures 2 and 3 are not sensitive to R.

3. Equation (10) and (11): what is the definition of m and what is its typical value?

Response: Thanks for the comment. m is a soil property parameter in famous Van Genuchten model. It presents the effect of soil porosity and usually determined by experiment. We added the definition and value in lines 137: " m=0.274 is the soil property parameter presenting the effect of soil porosity".

4. Equation (12): is the calculation of theta and evaporation rate applicable only over the wet soil? If the fraction of dry soil is > 0, i.e.,f_dust in Equation (1), will the theta and evaporation rate be calculated for that layer and how?

Response: Thanks for the comment. Theta and evaporation rate are calculated over the whole soil during dust emission event, even when the fraction of dry soil is > 0.

5. Section 2.4: a flow diagram would help illustrate the procedure. Lots of the detail about the model experiment are omitted. As mentioned in the main comment above, without those detail it is difficult to determine whether the results are reasonable. For instance, what is the initial soil moisture profile used? Is it representative for farmland, which seems to be the land surface type of interest? The model domain is unclear: is it a 1-D or 3-D model? What is the model horizonal and vertical resolution? Are there any horizonal variability in the initial conditions of soil moisture content and surface winds?

Response: Thanks for the comment. We added a flow diagram to help illustrate the procedure. In addition, 0.15 is set as an initial soil moisture in the whole soil to present a sufficient water condition after rainfall or irrigation, this is a vertical 1D model and the whole thickness of soil is 1m and the grid size is 1mm. Our dust emission model is established to simulate bare farmland condition, which horizonal variabilities in the initial conditions of soil moisture content and surface winds are not principal influence factors. More details please look at section 2.4. The calculation procedures can be seen in section 2.4, we have modified this part and add a flow diagram to make it easier for readers to understand in lines 163-165.

Figure 2. The flow diagram for dust emission model considering aerodynamic entrainment and surface renewal processes in a single time step.

6. Figure 3: there is no black lines plotted in any of the panels (a)-(c). During the first hour when $H_d > 0$, why the soil moisture remains constant but there is a slow increase in evaporation rate? is the stepwise increase in evaporation rate and soil moisture related to the initial soil moisture profile assumed?

Response: Thanks for the comment. We corrected the title of Fig. 3 in lines 203-205: "Figure 4: Temporal changes for evaporation and soil structure with different friction

velocity u_*: (a) u_*=0.4m/s; (b) u_*=0.45m/s; (c) u_*=0.5m/s. Green lines are dry soil layer thicknesses; blue lines are the evaporation rates; pink lines are the soil moisture on wet layer surface, which determine the evaporation rates.". It can be seen from Eq. 9 and Eq. 12 that the evaporation rate is higher with lower dry soil layer thicknesses, and the soil surface moisture becomes larger when underground water is enough. So, there is a slow increase in evaporation rate with the decreasing dry soil layer thicknesses. The initial soil moisture profile affects the initial values of evaporation rate and soil surface moisture, but the wind velocity is a more important factor determined the increasing rate.

7. Figure 4: in order to attribute the dust emission flux to a primary mechanism, it would make sense to plot the contribution of aerodynamic entrainment separately from that due to saltation transport. Sensitivity studies of other important parameters in the model such as soil moisture profile and surface air temperature/humidity would help in strengthening the findings from the model simulations.

Response: Thanks for the suggestions. We constructed more sensitivity tests on u*, specific humidity and initial soil moisture content in lines 220-222.

Figure 5: Sensitivity of dust emission flux F to friction velocity, specific humidity and initial soil moisture content. Three main phases in dust emission process: (i) aerodynamic entrainment is the primary mechanism in first phase, and the dust emission rate decreases rapidly in a few minutes, (ii) saltation transport is the main mechanism in the second phase, and the dust emission rate maintains at a relatively high level, (iii) soil moisture becomes the dominating limit factor in the third phase, and forms little dust emission.

. . . Tests are performed to investigate the dependency of dust emission F on friction velocity, specific humidity and initial soil moisture content. For constant friction velocity and initial soil moisture content, F has a small difference with large specific humidity, and clearly increases for in small specific humidity. Figure 5c also shows that final dust

emission rate F is insensitivity with initial soil moisture content.

8. Section 3.4: this model evaluation section needs to be expanded. As mentioned in the main comment, it is unclear if the model configuration is comparable to the experimental conditions such as soil type, moisture content and profile. More quantitative analysis of the model-data differences is needed, for instance, in terms of RMSE, correlation, or other statistical measures. The impact due to Surface Renewal and Evaporation (SRE) is visible only after 6 hours; however, this seems to be inconsistent with the model simulations in Section 3.1-3.3, which show that the SRE effects occur within âĹij 1hr. Why is that? Even after including the SRE effect, the differences between the model and observations are still quite large. It is not very convincing that the developed model captures the time evolution of the dust emission fluxes effectively in this case study. Furthermore, to attribute the changes of dust emission to a specific process: aerodynamic entrainment of free dust or saltation transport due to wind erosion, it is necessary to decompose the predicted dust emission fluxes by process.

Response: Thanks for the comment. In this version, we added some field experiment results and recalculated the model with consideration of the experimental conditions such as soil type, moisture content and profile in lines 254-257: "The model is calibrated and validated with field data from a sand storm monitoring station in the Horqin Sandy Land in China in 2011 (Li et al., 2014). The Horqin station has a 20 m observational tower, and the observations included wind speed at heights of 2, 4, 16, and 20 m; soil moisture at depths of 5, 20, and 50 cm; dust (particulate matter 10 (PM10)) concentration at heights of 3 and 18 m.". In fact, the third phase, which surface renewal caused by soil moisture, occurs after several hours in general due to the erosion process of dry soil layer. In addition, we added some field experiment results, which show the change of dust emission flux with time and the significant influence of surface renewal process in fig. 6:

Figure 6: (left) Time series of observed and modeled dust emission flux. The time is given in observation days (local time). Green triangles are wind velocity data measured

at the height of 2 m; red circles are the measured air dust emission rate. Black solid lines are the simulated dust emission flux considering surface renewal; black dotted lines are the cases without considering surface renewal. (right) Corresponding modeled versus observed fluxes for determination.

The model is calibrated and validated with field data from a sand storm monitoring station in the Horqin Sandy Land in China in 2011 (Li et al., 2014). The Horqin station has a 20 m observational tower, and the observations included wind speed at heights of 2, 4, 16, and 20 m; soil moisture at depths of 5, 20, and 50 cm; dust (particulate matter 10 (PM10)) concentration at heights of 3 and 18 m. Figure 6 shows the time series and scatterplots of the observations and the model results for four of the Horqin cases. At Horqin, fluxes of dust particles with diameters < 10 $\mu$m were estimated from the PM10 concentration profile measurements. As seen, there is a good agreement between the model predictions and observations and the temporal evolutions match well. For the three cases shown, the coefficient of determination, r2, is the lowest for the case of 2 May 2011 with r2 = 0.85 and the highest for the case of 19 May 2011 with r2 = 0.92. In the former case, the low r2 is caused by the poor model-observation agreement at about 16:00. For the remaining time, the predictions and observations differ only slightly in magnitude. In the latter case, the temporal evolution is well reproduced by the model with only slight discrepancies at about 14:00 and 18:00. Overall, in the four cases, the model predictions and observations agree with regard to onset and cessation as well as overall characteristics. Especially, in the latest case, the dust emission flux decreases obviously after 20:00 even though the wind velocity increases slightly, which indicates that u*t increases due to surface renewal process. As a contrast, the simulated dust emission flux without considering surface renewal increases with the wind velocity and is contrary to the observed dust flux due to the traditional models can't presents the change of soil property and u*.

9. Page 9, lines 14-15: dust emission caused by aerodynamic entrainment has been demonstrated in a number of previous studies such as Klose and Shao (2012) and

Zhang et al. (2016). For the statement "this model simulated the dust emission process caused by aerodynamic entrainment in nature for the first time", clarification about how this study is different from previous studies on this process is needed.

Response: Thanks for the comment. This is an unclear statement and we have modified this part in lines 227-243: "Fig. 5 shows different phases in the dynamic dust emission process. During the dust dynamic emission, the dust emission rate curve under different wind velocities showed a similar change trend, which could be divided into three main emission phases. The first phase was supplied by free fine dust mainly and aerodynamic entrainment emission was the primary mechanism. Due to the smaller grain size of free dust and the lower cohesive forces reduced by soil aggregates, the dust emission rate was very high in this phase. However, because the uneven distribution of free dust content in the vertical direction, the dust emission rate in this phase was decreased rapidly with time, reflecting the supply limitation of free dust. Klose and Shao (2012) study the aerodynamic entrainment in the absence of saltation as large eddies intermittently produce strong shear stresses on the surface and entrain dust particles into the air, in which convective atmospheric condition is major influence factor rather than soil property. While the free dust layer was consumed by wind erosion, saltation transport became the main mechanism in this phase. Because dust emission from big grains was relatively high and erosion processes were restrained accordingly in this phase, the dust emission rates were decreased significantly compared with that in the first phase. Therefore, the thickness of dry soil layer was main limiting factor of the dust emission in this phase. After the dry soil layer disappeared, the dust emission turned into the third phase, in which wet soil was the limit factor and saltation transport was the main mechanism. The existence of water between the soil grains hindered the releasing process of wind erosion and further reduced the dust emission rate. In this phase, soil moisture content became the main limiting factor of dust emission rate. Zhang et al. (2016) study the first two phases in wind tunnel experiments, but the third phase is difficult be rebuilt due to the time limit for wind tunnel operation.".

Overall, the manuscript is an interesting modeling study of dust emission processes based on the theoretical understanding. However, it requires significant improvement and justification in model description and evaluation, in order to support the findings of their model simulations.

Response: Thanks for the positive comments again, we have improved the quality of whole manuscript according to your suggestions and expect to hear more comments and suggestions from you.

[Figure]

**Fig. 1.**

[Figure]

**Fig. 2.**

[Figure]

**Fig. 3.**

[Figure]

**Fig. 4.**

[Figure]

**Fig. 5.**

[Figure]

**Fig. 6.**

---

## Author Comment (AC2) · 16 Mar 2021

Zhang et al., (2016) had put forth the renewal mechanism of fine particles in a soil's top layer, which they think is critical to sustaining dust emission. The work Dun and Huang presented here clearly attempts to build on that study simulating the dust emission process in farmland using a dust emission model with combined aerodynamic entrainment and surface renewal mechanisms previously proposed. They are trying to show that their model is effective to predict dust emission in farmland. In general, however, I think the performed approach and methodology are subject to major deficiencies, and the results are questionable. In many places, the statements drawn by the authors

lack sufficient evidence: the readers would appreciate it if the authors could explain some crucial aspects in detail. Some sections also needed to be restructured. So, I regret that I am unable to recommend publication of this manuscript in its present form in ACP. P – Page; L – Line (please use continuous line numbering instead of restarting numbering on every page. The current line numbering makes the review process painful!)

Response: Thanks for the comments and suggestions, we had provided more data and explanations, improved the quality of whole manuscript, and used continuous line numbering instead of restarting numbering on every page in the revised manuscript.

1. As the authors themselves pointed out, the simulated dust emission rate only slightly differs between including and excluding the aerodynamic entrainment and surface renewal mechanisms. I cannot find clear evidence supporting the main conclusion of this study. I have no idea based on what the authors concluded that "the model is an effective method to predict the dust emission rate". I encourage the authors to try to improve the model results, or, if that proves impossible, then learn why the model is not working and write a thoughtful and candid report characterizing the issues and the lessons learned by the attempt. But currently, I am really struggling to find out the scientific merit of this work.

Response: Thanks for the useful comment and suggestion. Generally, in dust emission models, the $u\_*t$ is considered to be constant during a dust erosion event. However, it has been found recently that, during an erosion event, the surface renewal process takes place and affects the dust emission by changing the soil particle distribution and soil moisture, and finally resulting in the change of $u\_*t$ (Li and Zhang, 2014; Zhang et al., 2016). Such phenomenon is common in natural, but no attempts have been made to model this process in physical sense coupling dust emission and surface renewal (Cornelis and Gabriels, 2010), which leads to an underestimation of simulated dust emission (Bergametti et al., 2016; Xin and Sokolik, 2015). This work is an attempt to improve the dust emission prediction, and the model results do match the observations.

According to the suggestions, we added Fig. 6 and some sentences to clearly explain the main conclusion of this study in lines 252-273.

Figure 6: (left) Time series of observed and modeled dust emission flux. The time is given in observation days (local time). Green triangles are wind velocity data measured at the height of 2 m; red circles are the measured air dust emission rate. Black solid lines are the simulated dust emission flux considering surface renewal; black dotted lines are the cases without considering surface renewal. (right) Corresponding modeled versus observed fluxes for determination.

The model is calibrated and validated with field data from a sand storm monitoring station in the Horqin Sandy Land in China in 2011 (Li et al., 2014). The Horqin station has a 20 m observational tower, and the observations included wind speed at heights of 2, 4, 16, and 20 m; soil moisture at depths of 5, 20, and 50 cm; dust (particulate matter 10 (PM10)) concentration at heights of 3 and 18 m. Figure 6 shows the time series and scatterplots of the observations and the model results for four of the Horqin cases. At Horqin, fluxes of dust particles with diameters $< 10 \ \mu$m were estimated from the PM10 concentration profile measurements. As seen, there is a good agreement between the model predictions and observations and the temporal evolutions match well. For the three cases shown, the coefficient of determination, r2, is the lowest for the case of 2 May 2011 with r2 = 0.85 and the highest for the case of 19 May 2011 with r2 = 0.92. In the former case, the low r2 is caused by the poor model-observation agreement at about 16:00. For the remaining time, the predictions and observations differ only slightly in magnitude. In the latter case, the temporal evolution is well reproduced by the model with only slight discrepancies at about 14:00 and 18:00. Overall, in the four cases, the model predictions and observations agree with regard to onset and cessation as well as overall characteristics. Especially, in the latest case, the dust emission flux decreases obviously after 20:00 even though the wind velocity increases slightly, which indicates that u*t increases due to surface renewal process. As a contrast, the simulated dust emission flux without considering surface

renewal increases with the wind velocity and is contrary to the observed dust flux due to the traditional models can't presents the change of soil property and u*.

2. Also, the evaluation of model performance (Section 3.4) relies on only one dust event, and there is no detailed quantitative analysis of the modeled and experimental data. Strictly speaking, a more intensive evaluation is required to put the conclusion on a more solid statistical basis. Event for comparison between the simulated and measured dust flux at the current level, the discussion seems somewhat subjective. The authors divided the field dust event process into three main phases without any justification. What kind of data or sensitivity study is there showing that this kind of phase division is reasonable? How did the authors distinguish the contribution of the dust emission from different mechanisms in each phase? How did they attribute the primary emission mechanism in the first phase to aerodynamic entrainment? Besides, the statement in the model evaluation section is a little bit not more candid. I would not say that the dust emission rate in the first phase, according to the authors' division, is high (the"relatively high" is vague): it is much lower than peak values registered in the second phase. In that short section, the authors mentioned the dust concentration twice. But I did not see any dust concentration data presented in the manuscript for this event to support those statements.

Response: Thanks for the suggestions. According to the experiments by Zhang et al. (2016), aerodynamic entrainment is highly effective if dust supply is unlimited, as in the first 2–3 min. While aerodynamic entrainment is suppressed by dust supply limits, surface renewal through the motion of surface particles appears to be an effective pathway to remove the supply limit. So, two phases are divided by from aerodynamic entrainment fine free dust supply is unlimited. In addition, from the field experiments reported by Li and Zhang (2014), the u*t will increase significantly and weaken the dust emission flux during a long-time dust emission event. They suppose that the amount of soil particles available for saltation is reduced due to the increasing soil moisture. We therefore take this phenomenon as the third phase. In this version, we added some

field experiment results, which show the change of dust emission flux with time and the significant influence of surface renewal process. For more details please see the response to question 1.

3. Also related to the field dust emission event. The authors should have to state where they get the data and how the experiment was set up to obtain the wind velocity and dust emission flux.

Response: Thanks for the suggestion. According to the suggestion, we had added some sentences in lines 258-261 of the revised manuscript as: "The model is calibrated and validated with field data from a sand storm monitoring station in the Horqin Sandy Land in China in 2011 (Li et al., 2014). The Horqin station has a 20 m observational tower, and the observations included wind speed at heights of 2, 4, 16, and 20 m; soil moisture at depths of 5, 20, and 50 cm; dust (particulate matter 10 (PM10)) concentration at heights of 3 and 18 m.".

4. The "farmland" only appears in the title and the abstract but is not mentioned anywhere else, which looks weird. The authors need to introduce somewhere in the manuscript the unique property of farmland surface from the dust emission perspective and make it clear why the model presented here is suitable for use to model the dust emission in farmland. It would be more interesting if the authors could quantitatively quantify how big the difference would be on the dust emission rate with and without the surface renew by soil moisture.

Response: Thanks for the suggestions. According to the suggestions, we added some sentences in lines 54-55 of the revised manuscript as: "Our dust emission model is established to simulate bare farmland condition, which the soil remains good grain size distribution and without crust covering due to the soil scarification and usually has adequate underground water supply." In addition, we quantified the difference with and without the surface renew by soil moisture in Fig. 6. More details please look at the response to question 1.

5. The calculation procedure is not very clear to me. I think there would be an update of theta after obtaining the evaporate rate. So, the theta in Eq. 12 is actually at a time step right before the current one. The authors may want to clarify the different time steps the theta is at in the equations to avoid any possible confusion. A flow chart would be helpful too.

Response: Thanks for the comment. According to the suggestions, we added the boundary conditions in Equation 13b, and presented the connection between theta and evaporate rate. In this way, the update of theta after obtaining the evaporate rate can be seen. In addition, we added a flow chart for calculation procedure to help understanding in lines 163-165.

Figure 2. The flow diagram for dust emission model considering aerodynamic entrainment and surface renewal processes in a single time step.

6. I would encourage the authors to construct more sensitivity tests on some key parameters that control the soil moisture prediction to see quantitively how they affect the dust emission rate in farmland.

Response: Thanks for the suggestion, we constructed more sensitivity tests on u_*, specific humidity and initial soil moisture content in lines 205-222.

Figure 5: Sensitivity of dust emission flux F to friction velocity, specific humidity and initial soil moisture content. Three main phases in dust emission process: (i) aerodynamic entrainment is the primary mechanism in first phase, and the dust emission rate decreases rapidly in a few minutes, (ii) saltation transport is the main mechanism in the second phase, and the dust emission rate maintains at a relatively high level, (iii) soil moisture becomes the dominating limit factor in the third phase, and forms little dust emission.

Fig. 5 shows different phases in the dynamic dust emission process. During the dust dynamic emission, the dust emission rate curve under different wind velocities showed

a similar change trend, which could be divided into three main emission phases. The first phase was supplied by free fine dust mainly and aerodynamic entrainment emission was the primary mechanism. Due to the smaller grain size of free dust and the lower cohesive forces reduced by soil aggregates, the dust emission rate was very high in this phase. However, because the uneven distribution of free dust content in the vertical direction, the dust emission rate in this phase was decreased rapidly with time, reflecting the supply limitation of free dust. While the free dust layer was consumed by wind erosion, saltation transport became the main mechanism in this phase. Because dust emission from big grains was relatively high and erosion processes were restrained accordingly in this phase, the dust emission rates were decreased significantly compared with that in the first phase. Therefore, the thickness of dry soil layer was main limiting factor of the dust emission in this phase. After the dry soil layer disappeared, the dust emission turned into the third phase, in which wet soil was the limit factor and saltation transport was the main mechanism. The existence of water between the soil grains hindered the releasing process of wind erosion and further reduced the dust emission rate. In this phase, soil moisture content became the main limiting factor of dust emission rate. Tests are performed to investigate the dependency of dust emission F on friction velocity, specific humidity and initial soil moisture content. For constant friction velocity and initial soil moisture content, F has a small difference with large specific humidity, and clearly increases for in small specific humidity. Figure 5c also shows that final dust emission rate F is insensitivity with initial soil moisture content.

7. Many variables are using in the equations without any definition. Values for constant parameters used in the model are also missing (please see detailed comments below). I would encourage the authors to specify those constants such that readers can tell if they are within the reasonable range and reproduce the results.

Response: We apologize for the mistakes and thanks for the suggestion, we carefully checked the equations added definitions and values for constant parameters in this

version, please look at the modified manuscript.

8. P1; L18: please give the size range for "fine particle".

Response: Thanks for the comment, we suppled the size range for "fine particle" in lines 18-19: "due to the loss of nutrient rich fine particles (d<60$\mu$m), coarsening of topsoil, decreasing of soil fertility and declining of land productivity (Shao, 2008; Mahowald, 2011; Huang et al., 2012)".

9. P1; L25-27: "Their results indicated that, in the initial phase of dust emission from a natural soil surface, aerodynamic entrainment should be the dominant mechanism and dust might be supplied by free grains exposed on soil surface." This statement seems not really correct. The aerodynamic entrainment could be crucial for dust emission only under certain circumstances. Here, I think that the authors exaggerated the importance of aerodynamic entrainment to dust emission.

Response: Thanks for the comment. In fact, field experiments and modeling perspectives find that the long-term contribution of recurrent aerodynamic dust entrainment plays an important role in dust production, particularly at low mean wind velocities (<7 m s^(-1)) (Ansmann et al., 2008; Macpherson et al., 2008; Shao, 2008; Sow et al., 2009; Allen et al., 2013; Klose et al., 2014), and Wind tunnel experiments of dust emissions from Zhang et al., (2016) also confirm Aerodynamic entrainment is highly effective in the initial phase of dust emission, if dust supply is unlimited. Of course, the statement in our manuscript is not very accurate, so we modify the statement in lines 23-28 as: "Recently, field experiments and modeling perspectives find that the long-term contribution of recurrent aerodynamic dust entrainment is substantial in nature (Ansmann et al., 2008; Macpherson et al., 2008; Shao, 2008; Sow et al., 2009; Allen et al., 2013; Klose et al., 2014), which leads Zhang et al. (2016) to studies on different phases in the dust emission from different soil surfaces. Their results indicated that, in the initial phase of dust emission from a natural soil surface, aerodynamic entrainment should be the dominant mechanism if dust supply is unlimited.".

10. P2; L7: how thick the topmost layer is defined?

Response: Thanks for the comment. We suppled the topmost layer information in lines 36-38: "Generally, in dust emission models, the soil moisture in whole topmost layer (at least 2 to 10 cm thick) from regional or global land surface is considered to be constant during a dust erosion event, which leads to an underestimation of simulated dust emission (Bergametti et al., 2016; Xin and Sokolik, 2015)".

11. P3; Section 2: the readers would appreciate a few sentences right after the section heading to explain how Section 2 is organized before diving into the subsections.

Response: Thanks for the suggestion, we added explanation in lines 54-60: "Our dust emission model is established to simulate bare farmland condition, which the soil remains good grain size distribution and without crust covering due to the soil scarification and usually has adequate underground water supply. We consider aerodynamic entrainment and surface renewal the main mechanisms. In the first component, we propose a simple and feasible scheme to calculate the amount of free grains exposed on soil surface, and offer an expression for dust emission for aerodynamic entrainment. The second component is to simulate the saltation process and surface renewal affected by soil moisture. In the third component, we detail how to predict the temporal soil moisture content using a soil moisture transport module. The calculation procedure and flow chart are presented in the last component".

12. P3; Eq. 1: consider adding a plot to Figure 1d to illustrate the vertical profile of the free fine dust coverage. Also, labeling the thickness of the free dust layer in Figure 1c would help readers get the point readily.

Response: Thanks for the suggestion, we modified Figure 1 in liens 46-52:

Figure 1. (a) The initial phase of dust emission due to aerodynamic entrainment and the second phase due to soil saltation (Zhang et al., 2016). (b) Soil moisture initials the erosion and dust emission rate in the third phase (Chen et al, 1996). (c) Illustration of

the soil structure: (i) free grains for aerodynamic entrainment, (ii) dry soil layer for soil saltation, and (iii) wet soil that inhibits the saltation and dust emission. (d) Soil moisture distribution along the depth and the residual soil moisture content ( ) is the threshold between dry and wet soil.

13. P3; Eq. 2: please introduce Fdust in the main text before showing this equation. Also, what is the value of "n" used in this study?

Response: Thanks for the comment. We added the introduction in line 72: "F_dust is the dust emission rate by aerodynamic entrainment from wind tunnel experiments (Zhang et al., 2016), ", and offered the values in line 76-77: " and n are the coefficient obtained from experiments.".

14. P3: does d in Eq. 3 refer to diameter? What is the size limit in the dust model?

Response: Thanks for the comment. We added the explanation for Eq. 3 refer to diameter in lines 85-88: "where d_s is the grain size of saltate soil ( $60\mu$m-$1000\mu$m), and c_0 as the grain terminal velocity, v_t the density ratio of grain to air, u_*wt is the threshold friction velocity considering soil moisture (Horikawa et al., 1983), u_*t is the threshold friction velocity (Shao and Lu, 2000),"

15. P3; L25: please introduce U*wt first and then U*t. Also, please define AN and gamma, and specify the constants used in the dust model.

Response: Thanks for the comment. We restructured the statement and added the explanation for constants in line 86-90: "u_*wt is the threshold friction velocity considering soil moisture (Horikawa et al., 1983), u_*t is the threshold friction velocity (Shao and Lu, 2000), and theta is the volume moisture content (%). For the constants, A_N being around 0.0123 and gamma being around $3*10^{-4}$ kg $s^{-2}$, rho_w=1000 kg $m^{-3}$ is the water density, pho=1.293 kg $m^{-3}$ is the air density, rho_s=1800 kg $m^{-3}$ is the dry bulk density of soil, rho_p=2650 kg $m^{-3}$ is the saltate particle density.".

16. P4; L1: please define rou w and rou s.

Response: Thanks for the comment. We added the explanation for constants in line 88-90: "For the constants, A_N being around 0.0123 and gamma being around $3*10^{-4}$ kg $s^{-2}$, rho_w=1000 kg $m^{-3}$ is the water density, pho=1.293 kg $m^{-3}$ is the air density, rho_s=1800 kg $m^{-3}$ is the dry bulk density of soil, rho_p=2650 kg $m^{-3}$ is the saltate particle density.".

17. P4; Eq. 4: what' the difference between ds and d? ds is the soil grain size, then what does d stand for? It seems to be a typo, as it does not make sense to have d on the right-hand side but ds on the other, and d only comes into play in this lognormal formula. Also, should specify how many modes ("N") and the lognormal distribution parameters used for calculation.

Response: Thanks for the comment. We modified the statement in lines 108-113: where N=4 is modes number of the superimposed lognormal distribution. D_j and $\sigma$_j are median mass grain size and geometric standard deviation of the jth grain size distribution mode. w_j is weight ratio of jth grain size distribution mode. The values of parameters are as given in Table 1.

18. P4; L7: considering deleting "in lognormal distribution". This term seems redundant, considering it had been mentioned in the sentence right before. No need to repeat the information.

Response: Thanks for the suggestion, we have deleted the statement "in lognormal distribution" in line 110.

19. P4; L16: Text starting from this line within this subsection is talking about soil moisture distribution, separated from the content of this section. Since the soil moisture distribution is strongly affected by the evaporation rate (in Section 2.4, the authors also cite Eqs. 7 and 8 together with Eq. 9 and so on), it would be better to put it in Section 2.3, where the authors detailed how to predict the soil moisture content. With this adjustment, the authors may want to change the subtitle of 2.3 to "Soil moisture distribution" or any other similar.

Response: Thanks for the suggestion, we restructured the manuscript and put soil moisture distribution part in Section 2.3

20. P4; L20: specify the value for Dv.

Response: Thanks for the comment. We added the value in lines 124: " $D\_v=10^{(-7)}$ $m^{(2)} s^{(-1)}$ is the diffusion coefficient".

21. P4; Eq. 8: define lambda and specify the value used.

Response: Thanks for the comment. We added the value in lines 127: "where lambda is the coefficient.".

22. P5; Eq. 10, 11, and 12: specify Ks, m, a, b, c, and d etc.

Response: Thanks for the comment. We added the explanation in lines 135-144: where $K\_s=5*10^{(-4)}$ m $s^{(-1)}$ is the saturated hydraulic conductivity of soil, theta is the relative saturation, m=0.274 is the soil property parameter presenting the effect of soil porosity, theta_r is the residual soil moisture content and the threshold between dry and wet soil, and theta_s is the saturated soil moisture content. Since the wind velocity u is the principal factor, the evaporation rate E can be expressed as (Schmutz and Namikas, 2018),

where E0 is the evaporation rate on water surface (Ta et al., 2009), e0 is the saturated vapor pressure in a thin layer above the pure water surface, e_z is the vapor pressure at height z above the water surface, delta is the thickness of dry soil and determined by theta_r, and the values of a, b, c, d are 2.513, -0.013, 20, 0.217, respectively.

23. P5; Section 2.4: please provide more details about each step: how the initial boundary conditions were set; what's the grid resolution; what's the time step etc. Please specify. Also, see major comments on step 4.

Response: Thanks for the comment. theta_0 is set as an initial soil moisture in the whole soil to present a sufficient water condition after rainfall or irrigation, the grid size

is 1mm and time step is set as 1s. More details please look at section 2.4 and the response to major comments on step 4.

24. P5; L24-26: is there any cause-and-effect relationship between the two sentences? Please explain.

Response: Thanks for the comment. It is an unclear statement, and we modified this part in lines 178-182: "Typically, before a dust emission or wind erosion event, a continuous soil drying process usually already exits to increase its erodibility (Webb and Strong, 2011). We calculated a 10-day evaporation process without wind from a soil with a moisture content of 0.025, and rebuilt the erodible soil structure containing dry layer and wet layer in nature.".

25. P5; L26 to P6; L2: sentence difficult to follow. Why a 10-day evaporation process? Is the "soil initial condition" referring to the one right after the 10-day evaporation finished? But under which friction velocity was used as you have three in Fig. 2abc?

Response: Thanks for the comment. According to the work by Song et al. (2018), dry soil layer appears and the thickness remains stable after a 10-day evaporation process. The 10-day evaporation process has no effect of wind, and the "soil initial condition" indeed referring to the one right after the 10-day evaporation finished. When we begin to simulate the dust emission, three different friction velocities u_*: 0.4m/s, 0.45m/s, and 0.5m/s are used in section 3, including Fig. 2abc.

26. P6; L20-21: unclear sentence. What does it mean when saying that the "erosion effect on dry soil layer could hardly be improved"?

Response: Thanks for the comment. It is an unclear statement, and we modified this part in lines 199-200: "With a high wind velocity, dry soil layer was easy eroded and denudated soon, while the ability of erosion on wet soil layer still had great potential.".

27. P6; Fig. 2: why in the first 0.25 hr, the surface position with U_*=0.5 m/s is higher than with U_*=0.45 m/s? Why in the first hour the soil moisture content at the newly

exposed surface with U_*=0.5 m/s is higher than with U_*=0.45 m/s, even though the surface position is comparable between the two cases? Another interesting but missing point is that increasing U* from 0.45 to 0.5 m/s did not lower the surface position in the first 0.5 and 0.75 hrs as much as increasing U* from 0.40 to 0.45 m/s.

Response: Thanks for the comment and suggestion. 1) In fact, in the first 0.25 hr, the surface position with u_*=0.5m/s (-6.1 mm) is lower than with u_*=0.45m/s (-5.8 mm), because dry soil layer is eroded faster with higher wind velocity. 2) It can be seen from Eq. 9 and Eq. 12 that the evaporation rate is higher with lower dry soil layer thicknesses, and the soil surface moisture becomes larger when underground water is enough. 3) In larger wind velocity cases (U_*=0.45 m/s and U_*=0. 5 m/s), dry soil layer is eroded away within 0.75 hrs and the erosion for wet layer is weak. So, dry soil layer is mainly eroded when U_* increases from 0.40 to 0.45 m/s in the first 0.5 and 0.75 hrs, but wet soil layer is mainly eroded when U_* increases from 0.45 to 0.5 m/s at last and the erosion velocity is weak.

28. P7; Fig. 3: I think it could be interesting to also show the total soil thickness. Fig. 3s: I did not see black lines.

Response: Thanks for the comment. We corrected the title of Fig. 3 in lines 203-205: "Figure 4: Temporal changes for evaporation and soil structure with different friction velocity u_*: (a) u_*=0.4m/s; (b) u_*=0.45m/s; (c) u_*=0.5m/s. Green lines are dry soil layer thicknesses; blue lines are the evaporation rates; pink lines are the soil moisture on wet layer surface, which determine the evaporation rates.". More details please look at the manuscript.
* * *
[Figure]

**Fig. 1.**

[Figure]

**Fig. 2.**

[Figure]

**Fig. 3.**

[Figure]

**Fig. 4.**

[Figure]

**Fig. 5.**

[Figure]

**Fig. 6.**